# LONG-TAILED RECOGNITION ON BINARY NETWORKS BY CALIBRATING A PRE-TRAINED MODEL

## ABSTRACT

Deploying deep models in real-world scenarios entails a number of challenges, including computational efficiency and real-world (*e.g.*, long-tailed) data distributions. We address the combined challenge of learning long-tailed distributions using highly resource-efficient binary neural networks as backbones. Specifically, we propose a calibrate-and-distill framework that uses off-the-shelf pretrained full-precision models trained on balanced datasets to use as teachers for distillation when learning binary networks on long-tailed datasets. To better generalize to various datasets, we further propose a novel adversarial balancing among the terms in the objective function and an efficient multiresolution learning scheme. We conducted the largest empirical study in the literature using 15 datasets, including newly derived long-tailed datasets from existing balanced datasets, and show that our proposed method outperforms prior art by large margins ($> 14.33\%$ on average).

## 1 INTRODUCTION

Witnessing the huge success of large-scale deep neural models including DinoV2 Oquab et al. (2023), MAE He et al. (2022), DALL·E 3, DALL·E 2 (Ramesh et al., 2022) and ChatGPT (Ouyang et al., 2022), we also observe surging demands to deploy various high-performance deep learning models to production. Unfortunately, the development of computationally efficient models has not yet taken into account the need to reflect real-world data characteristics, such as long tails in the training data (He & Garcia, 2009). As these models are developed with little consideration of long-tailed data, they exhibit unsatisfactory accuracy when deployed. On the other hand, current LT recognition methods (Cui et al., 2021; He et al., 2021; Zhong et al., 2021) largely assume sufficient computing resources and are designed to work with a large number of full precision parameters, *i.e.*, floating point (FP) weights.

As binary networks are at the extreme end of resource-efficient models (Rastegari et al., 2016), the long-tailed recognition performance using the 1-bit networks would roughly correspond to the 'worst-case scenario' for resource constrained LT. To show the achievable accuracy bound with extremely resource-efficient neural network models, we choose to benchmark and develop long-tailed recognition methods using binary networks as a challenging reference.

Inspired by LoRA (Hu et al., 2021) and LLaMA-Adapterv1 (Zhang et al., 2023) and -v2 (Gao et al., 2023), we hypothesize that a sufficiently large, *single* full precision pretrained model trained on non LT data, despite the lack of guaranteed domain overlap with the target LT data, may be adapted to *multiple* target LT datasets which can then be utilized as distillation teachers for training binary networks. Specifically, to adapt to the various target LT data, we attach a learnable classifier per dataset to the pretrained model that is trained to use the pre-trained representations to solve the LT recognition task, which we call '*calibration*.' We then use the calibrated network as distillation teachers to learn binary networks on the LT data with additional feature distillation loss.

Unfortunately, the efficacy of calibration would vary from dataset to dataset. Tuning the balancing hyperparamters in the distillation loss for each dataset may yield better performing binary networks but is less scalable. For a more generalizable solution agnostic to data distributions, we propose to adversarially learn the balancing hyperparameter that is jointly updated with the distillation loss.

Furthermore, to address the issue of varying image scales that arises when handling multiple datasets, we propose a computationally efficient multiscale training scheme (Jacobs et al., 1995; Rosenfeld, 2013) when calibrating the teacher. Unlike naïve multiscale training, whose computational cost linearly increases per each resolution, we propose a computationally efficient multiscale training scheme with negligible cost increases. Specifically, we use multiscale inputs in the calibration process, where most of the teacher network weights are not part of the backward computation graph. Then, we feed multi-scale inputs only to the *frozen* teacher in the distillation stage.

We name our method **Ca**librate a**n**d **D**istill: **L**ong-Tailed Recognition on the **E**dge or **CANDLE** for short. For extensive empirical validation, we conduct the largest empirical study in the literature using 15 datasets, where some are derived by undersampling existing non LT datasets and some are from existing LT benchmarks. In all experiments, our method outperforms prior arts by large margins (at least $+14.33\%$ on mean accuracy across the tested datasets).

**Contributions.** We summarize our contributions as follows:

- Addressing resource constrained long-tailed recognition with binary networks for the first time.
- Proposing a calibrate and distill framework that a sufficiently large FP pretrained model can be adapted to various target LT data to use as distillation teachers.
- Proposing an adversarial balancing scheme and efficient usage of multi resolution inputs for generalization to multiple data distributions.
- Empirically validating on 15 highly varied LT benchmarks for comprehensive empirical studies (the largest in the LT recognition literature).

## 2 RELATED WORK

**Long-tailed recognition.** We categorize deep learning models on LT distributions into several categories including class rebalancing (He & Garcia, 2009; Buda et al., 2018; Shen et al., 2016), logit adjustment (Cao et al., 2019; Cui et al., 2019; Tan et al., 2020), two stage methods – sequential learning of representation and classifier (Kang et al., 2019; Xiang et al., 2020; Ren et al., 2020; Tang et al., 2020; Li et al., 2020), and knowledge distillation (He et al., 2021; Wang et al., 2020a; Zhang et al., 2021a; Xiang et al., 2020).

Similar to our approach, He et al. (2021); Wang et al. (2020a); Zhang et al. (2021a); Xiang et al. (2020) use distillation to harness the additional supervision from the teacher for better LT recognition. But they train the teacher network from scratch on the LT data, which is less scalable than utilizing a single large pretrained model. Wang et al. (2020a); Zhou et al. (2020); Cui et al. (2021; 2022) propose to increase the model capacity which incurs additional costs. We note that directly increasing required computational resources is less realistic and resource efficient methods should be preferred.

We also note that while some recent works Foret et al. (2020); Na et al. (2022); Rangwani et al. (2022) propose methods that may be utilized to better train quantized models on LT data, actual training of quantized models on LT data is seldom explored.

**Binary networks.** Binary networks have attracted the attention of researchers as one of the promising solutions with extreme computational efficiency. Some approaches improve on the underlying architecture of binary networks (Rastegari et al., 2016; Lin et al., 2017; Liu et al., 2018; 2020; Bulat et al., 2021) or use neural architecture search to find better binary networks (Kim et al., 2020; Kim & Choi, 2021; Bulat et al., 2020). While most binary network research focuses on the supervised classification problem, other scenarios such as unsupervised representation learning (Shen et al., 2021; Kim & Choi, 2022) and object detection (Wang et al., 2020b) are also explored but less. Other approaches elaborate training schemes for the effective optimization of binary networks (Martinez et al., 2020; Han et al., 2020; Meng et al., 2020; Liu et al., 2021; Le et al., 2022). Unlike other approaches, some works explore using binary networks as backbones for previously unused scenarios such as unsupervised learning (Kim & Choi, 2022; Shen et al., 2021).

Recent successful attempts in this regime utilize a multistage training strategy in which less binarized versions of the network are trained on the target data beforehand. The trained network is later used as weight initialization and a distillation teacher when training the binary network (Martinez et al., 2020; Liu et al., 2020; Bulat et al., 2021; 2020; Kim et al., 2020; Kim & Choi, 2021; 2022). In contrast, we aim to adapt and use large pretrained models on potentially different data, which is much more scalable than the conventional multi-stage training strategy.

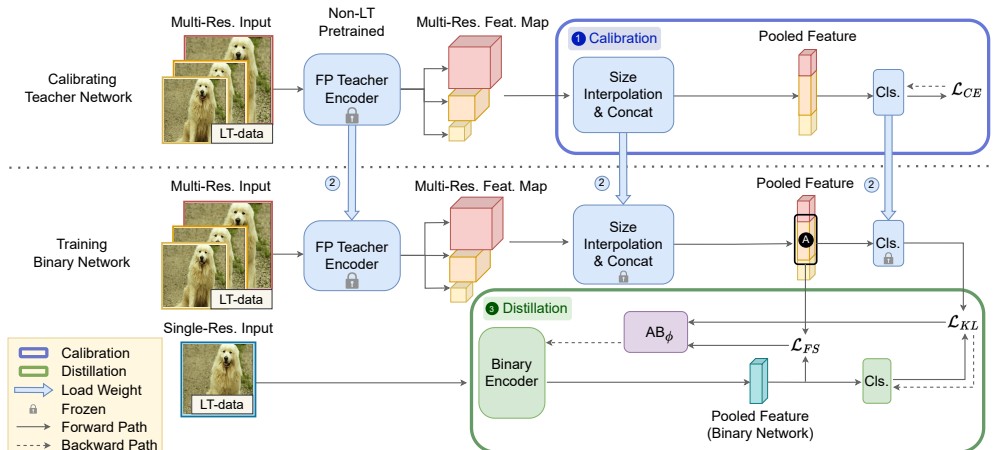

Figure 1: **Overview of the proposed CANDLE.** Learning progresses from ① to ③. The thick blue arrows (②) indicate copying the components of teacher network (used as frozen). In calibration, the pretraind FP teacher encoder is frozen (denoted by the lock icon) and the rest are calibrated with multi-resolution inputs (colored volumetric cubes for each resolution feature map, 'Multi-Res. Feat. Map'). In distillation, we compute $\mathcal{L}_{FS}$ by one of the calibrated teachers' multi-res. pooled feature (Ⓐ in the black rectangle) and compute $\mathcal{L}_{KL}$ by calibrated teachers' logit. The purple box ($AB_\phi$) is the adversarial balancing network (Sec. 3.2).

Moreover, long-tailed recognition using binary networks as backbones is seldom explored in the literature despite the efficiency that using binary networks can bring. Here, we aim to train resource efficient binary networks with accurate LT recognition capabilities to better prepare for LT recognition in real-world deployments.

# 3 APPROACH

Recently, Hu et al. (2021); Gao et al. (2023) show that adapting pretrained models to various target domains is effective for addressing downstream tasks. Given the long-tailedness of our target data, relying on the pretraining data overlapping tareget LT data is likely to be futile. Instead, we explore whether pretrained models can be utilized no matter what the overlap with the target LT data might be. In particular, we use pretrained (FP) models trained on large-scale public data (*e.g.*, ImageNet1K) which are adapted to target *long-tailed* data and then used as *distillation teachers* for training binary networks on the target LT data (see Sec. A for a brief preliminary on binary networks).

Additionally, since the adaptation efficacy may vary for different target LT data distributions which affects how useful the teacher supervision is, we propose a novel adversarial balancing of the feature distillation loss and the KL divergence loss terms that are used in the distillation. We further propose an efficient multiscale training to adapt the teachers to a potentially wide variety of input resolutions over multiple datasets.

## 3.1 CALIBRATE AND DISTILL

To utilize floating point (FP) pretrained models trained on non-LT data as teachers for training binary networks on the target LT data, we propose a simple 'Calibrate and Distill' scheme that adapts pretrained FP models from potentially different data than the target LT data.

Specifically, we attach a randomly initialized classifier to the feature extractor of the pretrained model. Then, only the classifier is trained using the LT-aware cross entropy loss $\mathcal{L}_{CE}$ (Zhong et al., 2021) on the target LT datasets during the calibration procedure, which is more efficient than training the entire teacher. We use the calibrated model as the distillation teacher and use the KL divergence ($\mathcal{L}_{KL}$) and feature similarity loss ($\mathcal{L}_{FS}$) between the calibrated teacher and the binary networks (Romero et al., 2014; Heo et al., 2019). The objective for learning binary networks with the calibrated teacher

is written as:

$$\min_{\theta} \mathbb{E}_{x \sim \mathcal{D}_{LT}}[(1 - \lambda) \cdot \mathcal{L}_{KL}(f^T(x), f^B_\theta(x)) + \lambda \cdot \mathcal{L}_{FS}(e^T(x), e^B_\theta(x))], \tag{1}$$

where $\mathcal{D}_{LT}$ is the target LT dataset and $f^T(\cdot)$ and $f^B_\theta(\cdot)$ are the calibrated teacher and binary networks, respectively. $e^T(\cdot), e^B_\theta(\cdot)$ are the teacher and binary encoders, *i.e.*, feature extractors, and $\mathcal{L}_{KL}(\cdot, \cdot), \mathcal{L}_{FS}(\cdot, \cdot)$ are the KL divergence and feature similarity defined by cosine distance (Chen et al., 2020; He et al., 2020), respectively. Note that unlike the common distillation methods that train the teachers for each task (He et al., 2021), our method is more scalable.

We also note that the calibration step allows us to utilize a pretrained model no matter how much the pretraining data overlaps with the target LT classes.

## 3.2 ADVERSARIAL BALANCING

For efficiency, we propose to adapt a single pretrained model for multiple target LT datasets while only retraining the classifier on each target data. Note that optimizing Eq. 1 using the same value of $\lambda$ for all target LT data is not optimal for all target LT datasets. To generalize our method to multiple datasets, we propose to parameterize and learn $\lambda$ by modifying Eq. 1.

For a given target LT dataset, for instance, if the feature vectors from the common feature extractor of teacher model provide better supervision via the feature distillation loss for the binary *encoder* than the dataset-specific teacher classifier via the KL divergence loss, we may want to emphasize $\mathcal{L}_{FS}$ by a larger $\lambda$. However, doing so will also affect how the binary *classifier* is trained as the parameters of the encoder and classifier share the same objective.

For more fine-grained control on these two terms, we propose to split the binary network's parameters $\theta$ into the encoder parameters ($\theta_e$) and classifier parameters ($\theta_c$), and optimize them separately as:

$$\begin{aligned} &\min_{\theta_c} \mathbb{E}_{x \sim \mathcal{D}_{LT}}[\mathcal{L}_{KL}(f^T(x), f^B_{\theta_c}(x))], \\ &\min_{\theta_e} \mathbb{E}_{x \sim \mathcal{D}_{LT}}[(1 - \lambda) \cdot \mathcal{L}_{KL}(f^T(x), f^B_{\theta_e}(x)) + \lambda \cdot \mathcal{L}_{FS}(e^T(x), e^B_{\theta_e}(x))]. \end{aligned} \tag{2}$$

**Adversarially learning the balancing ($\lambda$).** We then parameterize $\lambda$ by an MLP with learnable parameters $\phi$ to output a single scalar value $\lambda_\phi$ from the inputs $\mathcal{L}_{KL}$ and $\mathcal{L}_{FS}$ for the dataset specific balancing factor as:

$$\lambda_\phi = \text{AB}_\phi(\mathcal{L}_{KL}(x), \mathcal{L}_{FS}(x)), \tag{3}$$

where $\text{AB}_\phi(\cdot)$ denotes an adversarially balancing network that takes in $\mathcal{L}_{KL}(x)$ and $\mathcal{L}_{FS}(x)$, *i.e.*, two scalar values, and outputs a single scalar $\lambda_\phi$ (please see Sec. D for details).

Note that minimizing the loss with respect to $\phi$ results in a trivial solution of $\lambda_\phi = 0$ as the balancing coefficient for whichever loss function is larger, *i.e.*, 0 for $\mathcal{L}_{FS}$ in Eq. 2 if $\mathcal{L}_{FS} > \mathcal{L}_{KL}$. While this does *minimize* the loss, it also prevents the gradient flow from $\mathcal{L}_{FS}$ when optimizing with respect to $\theta$. Thus, we employ an *adversarial learning scheme* where $\lambda_\phi$ is learned to maximize the loss which is then minimized by optimizing $\theta$. We can rewrite our final adversarially learning objective as:

$$\begin{aligned} &\min_{\theta_c} \mathbb{E}_{x \sim \mathcal{D}_{LT}}[\mathcal{L}_{KL}(f^T(x), f^B_{\theta_c}(x))], \\ &\min_{\theta_e} \max_{\phi} \mathbb{E}_{x \sim \mathcal{D}_{LT}}[(1 - \lambda_\phi)\mathcal{L}_{KL}(f^T(x), f^B_{\theta_e}(x)) + \lambda_\phi \mathcal{L}_{FS}(e^T(x), e^B_{\theta_e}(x))]. \end{aligned} \tag{4}$$

We describe the algorithm for the adversarial loss balancing for a single data point in Alg. 1.

## 3.3 EFFICIENT MULTI-RESOLUTION LEARNING

As we use binary networks as backbones on LT data, we suffer more from the data scarcity apparent in the tail classes. Coupled with the fact that we want our model be robust for various image resolutions, multiscale training could be a favorable option (Rosenfeld, 2013). However, using multiscale input linearly increases the training time, as shown in Fig. 2 (Single-Res *vs*. Direct Multi-Res). Here, we propose a novel way to efficiently use multiresolution inputs for our 'Calibrate and Distill' framework such that there is negligible increase in training time (Single-Res *vs*. Our Multi-Res in Fig. 2).

---

**Algorithm 1** Adversarial Learning for $\lambda_\phi$

---

**Input**: binary network $f_\theta^B$, calibrated teacher $f^T$, binary encoder $e_{\theta_e}^B$, teacher encoder $e^T$, MLP $AB_\phi$, learning rate $\eta$, input data $x$
Compute teacher feature vector $e^T(x)$ and its logit $f^T(x)$
Compute binary feature vector $e_{\theta_e}^B(x)$ and its logit $f_\theta^B(x)$
Split $\theta$ into binary classifier weights $\theta_c$ and binary encoder weights $\theta_e$
$\mathcal{K} = \mathcal{L}_{KL}(f^T(x), f_\theta^B(x))$           // Compute KL divergence Loss (Eq. 2)
$\mathcal{F} = \mathcal{L}_{FS}(e^T(x), e_{\theta_e}^B(x))$           // Compute feature distillation loss (Eq. 2)
$\theta_c \leftarrow \theta_c - \eta \nabla_{\theta_c} \mathcal{K}$           // Update binary classifier
$\lambda_\phi = AB_\phi(\mathcal{K}, \mathcal{F})$           // Eq. 3
$\mathcal{L} = (1 - \lambda_\phi)\mathcal{K} + \lambda_\phi \mathcal{F}$           // Encoder loss
$\phi \leftarrow \phi + \eta \nabla_\phi \mathcal{L}$           // Adversarial learning of $\phi$ via maximization of the loss
$\theta_e \leftarrow \theta_e - \eta \nabla_{\theta_e} \mathcal{L}$           // Update binary encoder
**Output**: Updated $\theta_c, \theta_e, \phi$

---

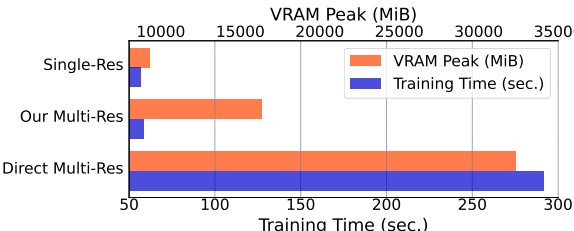

Figure 2: **Computational (blue bars) and memory (orange bars) costs comparison.** Between directly using multi-resolution inputs to our method of using them only for the teacher ('Our Multi-Res') in terms of VRAM peak (MiB) and 1 epoch training time (sec.).

As we use a teacher network for additional supervisory signals, we can encode the multi-resolution information in the calibration stage *before* distillation. We depict the process in the 'Training Teacher Network' part of Fig. 1. Since the calibration only trains the classifier (see Sec. 3.1), the usage of multi-resolution inputs is relatively inexpensive. During distillation, we only feed the multi-resolution inputs to the teacher.

Note that with the multi-resolution inputs, the feature similarity loss is calculated using only the channels that correspond to the input with the same resolution as that of the input given to the binary network as depicted by a black rectangle in the Fig. 1. Because the teacher is not part of the backward computational graph, training time negligibly increases, as shown in Fig. 2. In the figure, the 'VRAM peak' indicates the peak consumption of GPU memory during training. Our scheme does increase VRAM peak but is much more efficient than directly using the multi-resolution inputs to both teacher and binary networks.

## 4 EXPERIMENTS

**Experimental setup.** For empirical evaluations, we use 12 small-scale and 3 large-scale LT datasets. We use ReActNet-A (Liu et al., 2020) ($0.87 \times 10^8$ OPs) as binary backbone networks for all experiments, and otherwise mention it (*e.g.*, additional results using BNext-Tiny (Guo et al., 2022) in Sec. C). We use pretrained ResNet-152 (He et al., 2016) on ImageNet-1K for the small-scale and EfficientNetv2-M (Tan & Le, 2021) on ImageNet-21K for the large-scale LT datasets, which were chosen with consideration to the computational resources available at our end. We use $128 \times 128, 224 \times 224$ and $480 \times 480$ resolution images as multiscale inputs, following (Tan & Le, 2019).

Following (Liu et al., 2021), we use the Adam optimizer with zero weight decay for training binary networks. Further implementation details are stated in Sec. D. We use the authors' implementation of prior arts when available or re-implement by reproducing the reported results. More details on the compared prior arts are in Sec. E.

**Baselines.** We compare with $\tau$-norm (Kang et al., 2019), BALMS (Ren et al., 2020), PaCo (Cui et al., 2021), MiSLAS (Zhong et al., 2021), and DiVE (He et al., 2021).

**Extended benchmarks for long-tail recognition.** In addition to developing a method for LT recognition using binary networks, we also want to benchmark it with many different LT data distributions (Yang et al., 2022). To that end, we compile a large number of derived LT datasets from

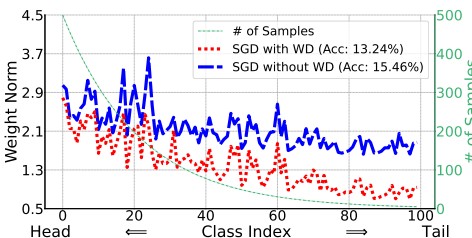

Figure 3: **Classifier weight norm of binary networks trained from scratch on CIFAR-100 (100) using SGD with or without weight decay.** We observe similar trends for the class weight norms as in (Kang et al., 2019; Alshammari et al., 2022) where the class weight norms become smaller for the tail classes. (See discussion in Sec. 4.1)

commonly used computer vision datasets along with the datasets often used in the LT literature; 12 small-scale and 3 large-scale datasets for evaluation.

We denote the imbalance ratio in parentheses after the dataset names for the small-scale datasets and add a postfix of '−LT' to each of the large-scale dataset names. More details on the 15 datasets are given in Sec. F. We will publicly release the code for CANDLE and the newly derived LT datasets to facilitate future research.

### 4.1 CLASSIFIER WEIGHT NORM ANALYSIS FOR BINARY NETWORKS IN LT DISTRIBUTION

One of the common problems in LT recognition is the data scarcity in tail classes, which causes the classifier weights to exhibit differing magnitudes across the head and tail classes(Kang et al., 2019; Alshammari et al., 2022; Wang et al., 2020a). Prior art has shown that these differences between classes in weight norms negatively impact the performance of LT recognition (Wang et al., 2020a). To gain further insight for the LT recognition performance with binary networks, we first investigate the classifier weights norms of trained binary networks, either from scratch or using our method.

We first visualize the classifier weight norms for binary networks in Fig. 3. While the weight norms are imbalanced similar to (Kang et al., 2019), there are other factors to consider. For instance, the accuracy of the trained binary networks are less than $16\%$, which is very low. The main reasons for the low accuracy is that we followed previous work (Kang et al., 2019; Zhou et al., 2020; Alshammari et al., 2022)'s training settings that use weight decay and the SGD optimizer. But they are not desirable choices; it is known that the Adam optimizer significantly outperforms the SGD optimizer and that weight decay is harmful when training binary networks (Han et al., 2020).

Thus, we repeat the analysis using SGD without weight decay (Fig. 3). Although accuracy is improved ($15.46\%$ compared to $13.24\%$), we believe that the binary network is still insufficiently trained. However, when we train with Adam, the accuracy for binary networks increases more than twice ($> 30.00\%$) and analysis is done using Adam without weight decay (Fig. 4-(a)), where the full precision version of the binary network architecture is also shown for comparison. Interestingly, when the model is trained from scratch, we observe that the classifier weight norms become larger as the classes are closer to the tail (*c.f.*, smaller in (Kang et al., 2019)).

We suspect that the different trend from (Kang et al., 2019) is due to the different optimization behavior of Adam compared to SGD (see Sec. B). Note that the class weight norms being larger for tail classes is also undesirable for LT as it would harm the accuracy of head classes instead (Kang et al., 2019). We also note that the average slope of the curve is sharper for binary networks. It implies that the degree to which the magnitudes of the classifier weights differ across classes could be higher for binary networks.

In contrast, when we apply the proposed CANDLE and conduct the same analysis (Fig. 4-(b)), the magnitudes of classifier weight norms of both binary and FP versions are substantially less different than in Fig. 4-(a). We attribute the reduced difference to the 'Calibrate and Distill' component of CANDLE that supplies additional supervisory signals, which could be beneficial for training capacity-limited binary networks on LT data.

### 4.2 LONG-TAILED RECOGNITION RESULTS

We present comparative results of our method with various state-of-the-art LT methods tuned for binary networks in Table 1 and 2. Head, median, and tail class accuracy improvements over the baseline in Table 3 is also shown. The best accuracy is in **bold** and the second best is underlined.

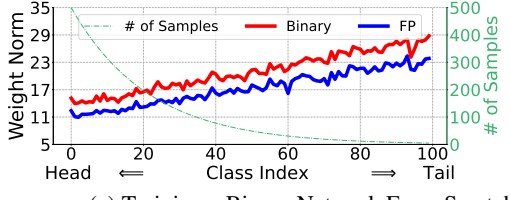 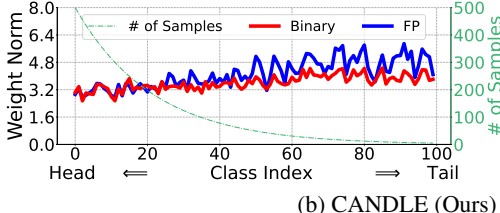

(a) Training a Binary Network From Scratch          (b) CANDLE (Ours)

Figure 4: **Classifier weight norms of binary and FP models trained with Adam without weight decay on CIFAR-100 (imbalance ratio: 100)**. (a) The classifier weight norms increase at the tail classes. When trained from scratch, binary networks show larger differences of weight norms than FP, as indicated by the larger slope of the curves. (b) Using CANDLE, the magnitudes of the weights of both binary and FP networks vary less, showing smaller differences across classes.

Table 1: **Average accuracy (%) of the proposed CANDLE and other methods on 12 small-scale datasets.** DiVE* is trained using the same teacher as ours. Our CANDLE outperforms state of the arts by large margins (at least +14.90% in average) across many different datasets. 'Imb. Ratio' refers to imbalance ratio between major and minor classes.

| Dataset (Imb. Ratio) | $\tau$-norm | BALMS | PaCo | MiSLAS | DiVE | DiVE* | **CANDLE** |
|---|---|---|---|---|---|---|---|
| Caltech101 (10) | 20.68 | 17.00 | 45.85 | 36.38 | 20.91 | 39.66 | **55.31** |
| CIFAR-10 (10) | 24.28 | 17.40 | 58.59 | 82.25 | 13.73 | 79.46 | **91.76** |
| CIFAR-10 (100) | 21.72 | 16.50 | 61.57 | 69.50 | 16.26 | 58.41 | **85.01** |
| CIFAR-100 (10) | 8.18 | 2.70 | 49.63 | 47.03 | 2.26 | 49.93 | **66.09** |
| CIFAR-100 (100) | 5.56 | 3.40 | 39.66 | 32.31 | 1.94 | 32.95 | **50.88** |
| CUB-200-2011 (10) | 8.52 | 1.30 | 15.41 | 14.74 | 4.09 | 12.82 | **42.94** |
| Stanford Dogs (10) | 13.93 | 9.80 | 34.04 | 29.73 | 6.72 | 29.21 | **58.79** |
| Stanford Cars (10) | 7.73 | 2.90 | 23.74 | 24.82 | 4.49 | 19.31 | **51.06** |
| DTD (10) | 16.86 | 14.10 | 36.65 | 36.54 | 12.71 | 30.37 | **38.56** |
| FGVC-Aircraft (10) | 7.71 | 4.30 | 15.12 | 22.08 | 1.95 | 14.19 | **39.78** |
| Flowers-102 (10) | 45.29 | 43.60 | 52.16 | 60.20 | 23.82 | 58.53 | **64.61** |
| Fruits-360 (100) | 85.82 | 97.30 | 99.23 | 99.49 | 50.87 | 99.71 | **100.00** |
| Average | 22.23 | 19.19 | 44.30 | 46.14 | 13.31 | 41.80 | **62.07** |

Table 2: **Average accuracy (%) of CANDLE and other methods on large-scale datasets.** † indicates that we only use single-resolution due to VRAM limits. DiVE* is trained using the same teacher network as ours. The proposed method outperforms other existing works in all three large-scale datasets with at least +7.93% margin in average.

| Dataset | $\tau$-norm | BALMS | PaCo | MiSLAS | DiVE | DiVE* | **CANDLE†** |
|---|---|---|---|---|---|---|---|
| Places-LT | 25.99 | 23.60 | 23.30 | 28.49 | 20.96 | 22.93 | **34.11** |
| ImageNet-LT | 30.59 | 33.00 | 34.58 | 34.71 | 31.21 | 30.86 | **49.10** |
| iNat-LT | 37.36 | 44.30 | 45.73 | 43.59 | 41.32 | 41.30 | **47.38** |
| Average | 31.31 | 33.63 | 34.54 | 35.60 | 31.16 | 31.70 | **43.53** |

**In small-scale datasets.** We summarize the results on the 12 small-scale datasets in Tab. 1 along with the mean accuracy over all 12 datasets. The mean accuracy over the 12 datasets suggests that CANDLE performs better than the prior arts by noticeable margins of at least +15.93%. More precisely, the proposed method outperforms all other existing works by large margins for the popular CIFAR-10 (10, 100) and CIFAR-100 (10, 100) datasets. In addition, our method exhibits superior performance in some of the newly added LT datasets with limited data such as CUB-200-2011 (10).

Surprisingly, DiVE (He et al., 2021) does not perform well on the small-scale datasets as the teacher network used in DiVE's distillation process, *e.g.*, BALMS (Ren et al., 2020) exhibits disappointing accuracy (∼ 26.65% mean acc.) in the first place. Hence, we also present results with DiVE* which uses the *same* teacher network as our method. We further discuss some of the other low performing methods (τ-norm (Kang et al., 2019) and BALMS) on the small-scale datasets in Sec. G.

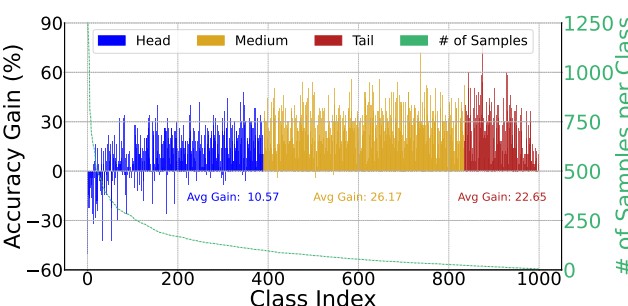

Figure 5: **Per-class accuracy gain of CANDLE over the baseline on ImageNet-LT**. Our method improves the accuracy at the tail classes by $+22.65\%$, showing its effectiveness for LT. Furthermore, the proposed method also shows gains in the accuracy at the head and medium classes by $+10.57\%$ and $+26.17\%$ respectively.

Interestingly, comparing DiVE and DiVE$^*$, we can see that using our calibrated teacher boosts the performance significantly. However, the performance difference between DiVE$^*$ and CANDLE $(+20.27\%)$ also indicates that our adversarial balancing (Sec. 3.2) and multi-resolution teacher (Sec. 3.3) is effective. Further comparison against the ensemble-based method, TADE (Zhang et al., 2021b), is also in Sec. I.

**In large-scale datasets.** Similar to Tab. 1, we compare our method to prior arts on the large-scale datasets in Tab. 2. While our multi-resolution teacher is far more efficient than directly using multi-resolution (Fig. 2), the VRAM peak still increases. Thus, we could not run the multi-resolution experiments with the large-scale datasets with our resources and only use single-resolution, indicated using † in Tab. 2. Still, even without using multi-resolution, the proposed method outperforms prior arts with binary networks on all three large-scale datasets. Looking at the mean accuracy, our method shows a margin of at least $+7.93\%$ when compared to the previous LT works.

**Analysis on head, median, and tail class accuracy.** In Fig. 5, we plot the per-class accuracy gain of CANDLE on ImageNet-LT against the baseline used in Table 3 to better understand at which classes the gains in mean accuracy of the proposed method are concentrated at. We color the head, median, tail classes by blue, gold and red, respectively for visual clarity. The gains in the tail class accuracy $(+22.65\%)$ and median class accuracy $(+26.17\%)$ are substantially higher than that of the head class accuracy $(+10.57\%)$.

This suggests that the the good performance of CANDLE in mean accuracy are mostly coming from solid improvements in tail and median classes. Note that the improvements of the proposed method in tail and median classes do not come at substantial costs in head class accuracy either. Additional per-class accuracy gains on Places-LT and iNat-LT can be found in Sec. J along with head, median, and tail class accuracy of CANDLE for all 15 datasets (small- and large-scale).

## 4.3 LEARNED $\lambda_\phi$ IN DIFFERENT DATASETS

To investigate the difference in balancing between $\mathcal{L}_{KL}$ and $\mathcal{L}_{FS}$ (Eq. 4) for different datasets, we plot the value of $\lambda_\phi$ vs. training iteration for each dataset in Fig. 6. Interestingly, we observe smooth transitions from initial values to 1 for all the datasets shown. The initial values are different because they depend on the different initial loss values for each dataset.

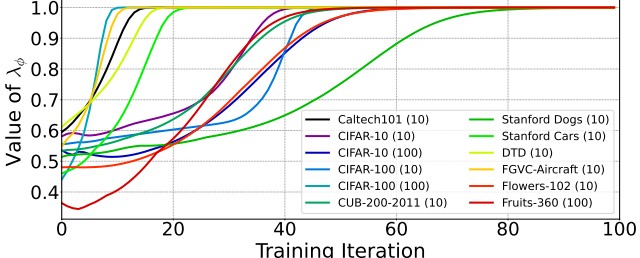

Figure 6: **Learned value of $\lambda_\phi$ on different datasets during training**. The convergence behavior of $\lambda_\phi$ to 1.0 differ for each dataset.

In addition, while the learned $\lambda_\phi$ converges to 1 (using only the $\mathcal{L}_{FS}$) in Eq. 4, the precise schedule is different per dataset. This indicates that the binary encoder is eventually trained with just the supervision from the FP teacher encoder, but the precise schedule varies from dataset to dataset.

The $+5.40\%$ gain in average accuracy in Tab. 3 of using adversarial balancing along with the different *learned* schedules of $\lambda_\phi$ in Fig. 6 imply that it is effective for LT recognition on a variety of datasets.

Table 3: **Ablation study of our CANDLE**. We present the mean accuracy for the 12 small-scale datasets. Using our 'Calibration & Distill' framework drastically improves the performance from the baseline of using the pretrained weights without calibration. Adversarial balancing and multi-resolution inputs also improve the mean accuracy substantially. C&D, ALB and MR refer to 'Calibrate and Distill', 'Adversarial Balancing' and 'Multi-resolution', respectively.

| Method | ① C&D | ② ALB. | ③ MR | Accuracy (%) | | | |
|---|---|---|---|---|---|---|---|
| | | | | Avg. | Head | Med. | Tail |
| Baseline | ✗ | ✗ | ✗ | 45.27 | 53.55 | 44.40 | 32.47 |
| ① | ✓ | ✗ | ✗ | 55.59 | 62.99 | 55.90 | 42.57 |
| ①+② | ✓ | ✓ | ✗ | 60.99 | 66.64 | 61.72 | 49.61 |
| ①+②+③ | ✓ | ✓ | ✓ | **62.07** | **71.22** | **63.33** | **50.09** |

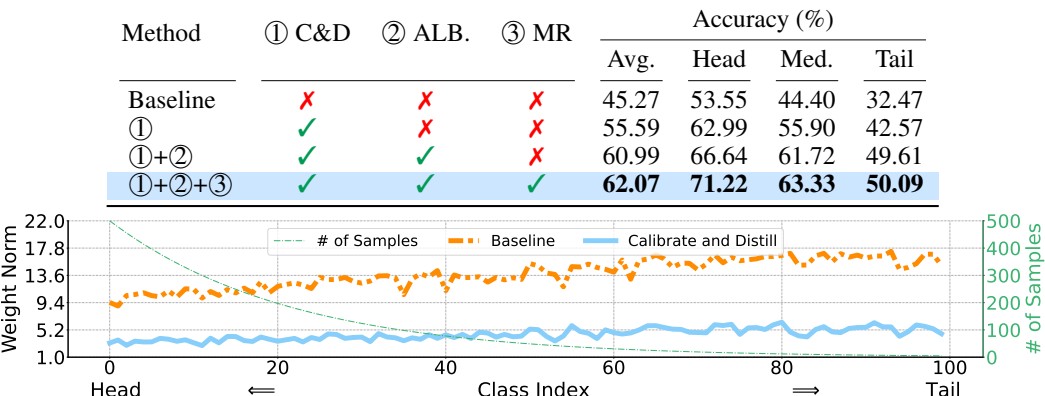

Figure 7: **Classifier weight norms of 'Calibrate and Distill' (CD) and 'Baseline' of using the pretrained weights without calibration on CIFAR-100.** (Imbalance ratio: 100) The CD maintains the magnitudes to be relatively similar for head to tail classes whereas the baseline does not, showing the effectiveness of the CD in reducing the differences of classifier weights across classes.

## 4.4 ABLATION STUDIES

We conduct ablation studies of our ① 'Calibrate & Distill', ② 'Adv. Learned Bal.', and ③ 'Multi-Res.' on the 12 small-scale datasets and report the mean average, head, medium, and tail class accuracy in Tab. 3. The baseline is set to using the uncalibrated teacher with feature distillation (Romero et al., 2014; Heo et al., 2019) (refer to Sec. 3.1 for detail).

Note that the baseline achieves $45.27\%$ mean average accuracy, only behind MiSLAS. All proposed components of CANDLE improve the mean average accuracy, *e.g.*, $+10.32\%$ for 'C & D', $+5.40\%$ for 'ALB.', and $+1.08\%$ for 'MR.'. Furthermore, the mean tail class accuracy also improves consistently with each component, indicating the effectiveness of the respective components for LT.

**Classifier Weight Norm Analysis.** Additionally, we conduct the classifier weight norms analysis (Sec. 4.1) for 'Baseline' and 'Calibrate & Distill' to investigate which improves the accuracy in Fig. 7. It shows that 'Calibrate & Distill' is much more effective in reducing the data imbalance problem in LT than the 'Baseline'; showing less fluctuated classifier weight norm across the classes. We show more detailed ablation study for each dataset in Sec. K for the space sake.

## 5 CONCLUSION

To develop efficient deep learning models for LT, we use binary networks as backbones for long-tailed recognition. We propose a simple 'Calibrate and Distill' framework where pretrained floating point models are calibrated to the target LT data to use as distillation teachers for learning binary networks. To better prepare the model for the wide variety of semantic domains and input resolutions present in the wild, we propose adversarial balancing and efficient usage of multi-resolution inputs.

We empirically validate the proposed CANDLE and other existing works on a total of 15 datasets, which is the largest benchmark in the literature. The proposed method achieves superior performance in all empirical validations with ablation studies showing clear benefits of each component of CANDLE despite its simplicity.

**Limitations and Future Work.** While our experiments mostly focused on using non-LT pretrained teacher models which are then calibrated for target LT data, it would be interesting to see if pretraining on a *large-scale* LT data in the first place will result in a better teacher.

ETHICS STATEMENT

This work aims to learn edge-compatible binary networks with long-tailed recognition capabilities. Thus, though there is no intent from the authors, the application of deep models to real-world scenarios with long-tail data distributions might become prevalent. This may expose the public to discrimination by the deployed deep models due to unsolved issues in deep learning such as model bias. We will take all available measures to prevent such outcomes as that is *not* our intention at all.

REPRODUCIBILITY STATEMENT

We take the reproducibility in deep learning very seriously and highlight some of the contents in the manuscript that might help in reproducing our work. First, we will release the code and newly derived datasets used in our experiments as mentioned in Sec. 4. Second, we include additional information regarding the extended long-tailed benchmark in Sec. F that may help with future work looking to use the datasets that were used in this work. Third, we include relevant implementation details in both Sec. 4 and Sec. D. Last, we present our final optimization objective (Eq. 4) and the overall view of our method (Fig. 1) with necessary details to reproduce the methodology.

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

APPENDIX

## A  A PRELIMINARY ON LEARNING BINARY NETWORKS (SEC. 3)

Let $\mathbf{W_F} \in \mathbb{R}^{C_i \times C_o \times k \times k}$ be the 32-bit floating point weights of a convolution layer where $C_i, C_o$ and $k$ represent the number of input channels, the number of output channels and the kernel size, respectively. The corresponding binarized weights are given by $\mathbf{W_B} = \mathrm{sign}(\mathbf{W_F})$, where the $\mathrm{sign}(\cdot)$ outputs $+1/-1$ depending on the sign of the input. The floating point input activation $\mathbf{A_F} \in \mathbb{R}^{C_i \times w \times h}$, where $w, h$ are the width and height of the input activation, can be similarly binarized to the binary input activation by $\mathbf{A_B} = \mathrm{sign}(\mathbf{A_F})$. A floating point scaling factor is also used, which can either be the L1-norm of $\mathbf{W_F}$ (Rastegari et al., 2016) or learned by back-propagation (Bulat & Tzimiropoulos, 2019).

The forward pass using convolutions is approximated as following

$$\mathbf{W_F} * \mathbf{A_F} \approx \alpha \odot (\mathbf{W_B} \oplus \mathbf{A_B}), \tag{5}$$

where $*, \odot, \oplus$ denote the floating point convolution operation, Hadamard product, and the binary XNOR convolution with popcount, respectively. STE (Courbariaux et al., 2016) is used for the weight update in the backward pass.

## B  DETAILS ON THE DIFFERENT TREND DUE TO ADAM COMPARED TO SGD (SEC. 4.1)

Using Adam results in the tail class weights having large norms which is different from using SGD, possibly due to how Adam adjusts the learning rate. Specifically, the Adam optimizer adjusts the learning rates such that parameters corresponding to frequently occurring head classes use relatively small learning rates and parameters corresponding to infrequent tail classes use relatively large learning rates. Thus, relatively larger learning rate are used in the update for tail classes, which may result in large class weight norms for tail classes.

## C  ADDITIONAL RESULTS USING BNEXT-TINY AS BINARY BACKBONE (SEC. 4)

We present additional results using our CANDLE with a different binary backbone, BNext-Tiny (Guo et al., 2022) , in Tab. 5 to show that improvements from backbone architectures could translate well to the proposed method as it is not specific to a single architecture. We chose BNext-Tiny as it has similar OPs count to ReActNet-A but performs better, as summarized in Tab. 4. As shown in Tab. 5, using a better backbone such as BNext-Tiny improves performance except for 2 datasets; Caltech101 (10) and CIFAR-100 (100). As such, the mean accuracy over the 12 datasets compared increases by $+1.18\%$. Thus, improvements from better binary backbones translate well with our method, implying that our method is not specific to a certain binary backbone architecture.

Table 4: Summary of the OPs (floating point + binary operations) count and ImageNet-1K classification of the two binary backbones used along with floating point (FP) ResNet-18 for reference. $\cdot^*$ indicates the performance including advanced training settings from (Guo et al., 2022).

| Backbone | OPs ($\times 10^8$) | ImageNet-1K Acc. (%) |
|---|---|---|
| ResNet-18 (FP) | 18.00 | 69.60 |
| ReActNet-A (Binary) | 0.87 | 69.40 |
| BNext-Tiny$^*$ (Binary) | 0.88 | 72.40 |

Table 5: Accuracy (%) of the proposed CANDLE with different binary backbone networks shown in Tab. 4.

| Datasets (Imb. Ratio) | **CANDLE** w/ ReActNet-A | **CANDLE** w/ BNext-Tiny |
|---|---|---|
| Caltech101 (10) | 55.31 | 50.35 |
| CIFAR-10 (10) | 91.76 | 92.29 |
| CIFAR-10 (100) | 85.01 | 85.33 |
| CIFAR-100 (10) | 66.09 | 66.10 |
| CIFAR-100 (100) | 50.88 | 48.33 |
| CUB-200-2011 (10) | 42.94 | 46.43 |
| Stanford Dogs (10) | 58.79 | 62.26 |
| Stanford Cars (10) | 51.06 | 54.63 |
| DTD (10) | 38.56 | 41.33 |
| FGVC-Aircraft (10) | 39.78 | 43.95 |
| Flowers-102 (10) | 64.61 | 68.04 |
| Fruits-360 (100) | 100.00 | 100.00 |
| Mean Acc. | 62.07 | 63.25 |

---

**Algorithm 2** Multi-resolution Teacher Forward

---

1: **Input**: calibrated teacher encoder $e^T$ and classifier $c^T$, input data $x$
2: $x_S, x_B, x_L = \text{Downsample}(x), \text{Identity}(x), \text{Upsample}(x)$     // Resize input to multiple resolutions
3: Teacher feature vectors: $e_S^T, e_B^T, e_L^T = e^T(x_S), e^T(x_B), e^T(x_L)$
4: $e_S^T, e_L^T = \text{Downsample}(e_S^T), \text{Upsample}(e_L^T)$
5: $l = c^T(\text{ChannelWiseConcat}([e_S^T, e_B^T, e_L^T]))$     // Compute classifier logits
6: **Output**: teacher feature vector $e_B^T$, teacher logits $l$

---

## D  FURTHER IMPLEMENTATION DETAILS (SEC. 4)

In the calibration stage, we used the class-balanced sampling and label-aware smoothing following MiSLAS (Zhong et al., 2021). We use a pretrained ResNet-152 (He et al., 2016) backbone for the ImageNet1K teacher and a pretrained EfficientNetv2-M (Tan & Le, 2021) backbone for the ImageNet21K teacher. In the distillation stage, we used ReActNet-A (Liu et al., 2020) as our binary backbone network. $\text{MLP}_\phi$ is an MLP with 3 hidden layers with the number of channels expanding by 16 per layer starting at 2 channels in the first layer. The $\text{MLP}_\phi$ takes two scalar values $\mathcal{L}_{KL}(x)$ and $\mathcal{L}_{FS}(x)$ turned into a 2-dimensional vector and outputs a single scalar value which is passed through a sigmoid function to be used as $\lambda_\phi$.

To process the multi resolution inputs to the teacher network, we use the size interpolation & concat module. The size interpolation & concat module is comprised of 1x1 convolutions that match the number of channels to that of the binary network and a nearest interpolation step where the different sized feature maps from the multi-resolution inputs are spatially resized. Once the feature maps from the different resolution inputs have the same spatial size and number of channels, they are channel-wise concatenated. A detailed algorithm of how multi-resolution inputs are used in the teacher forward is given in Alg. 2. Notice how only the teacher feature vector corresponding to the original resolution input are returned.

To optimize the min-max objective of Eq. 4, we perform 1 maximization step per every 1 minimization step. The binary networks are trained for 400 epochs with a batch size of 256. We use the cosine annealing scheduling for our learning rate scheduler, where the initial learning rate is set as 0.01 for most of the datasets. Although we did try different learning rates, we found the difference to be negligible.

We used one GPU for training on the 12 small-scale datasets and four GPUs for training on the 3 large-scale datasets.

Table 6: Dataset statistics of all the LT datasets used in our experiments. The dataset source denotes the original non-LT dataset from which the LT datasets were derived. The imbalance ratio for the large scale datasets are not explicitly calculated.

| Dataset Source | | Imbalance Ratio | # Train Samples | # Test Samples |
|---|---|---|---|---|
| | Caltech101 | 10 | 1151 | 6084 |
| | CIFAR-10 | 10 | 20431 | 10000 |
| | CIFAR-10 | 100 | 12406 | 10000 |
| | CIFAR-100 | 10 | 19573 | 10000 |
| | CIFAR-100 | 100 | 10847 | 10000 |
| | CUB-200-2011 | 10 | 2253 | 5794 |
| Small Scale | Stanford Dogs | 10 | 4646 | 8580 |
| | Stanford Cars | 10 | 3666 | 8041 |
| | DTD | 10 | 1457 | 1880 |
| | FGVC-Aircraft | 10 | 2564 | 3333 |
| | Flowers-102 | 10 | 753 | 1020 |
| | Fruits-360 | 100 | 1770 | 3110 |
| | Places-LT | N/A | 62500 | 7300 |
| Large Scale | ImageNet-LT | N/A | 115846 | 50000 |
| | iNat-LT | N/A | 437513 | 24426 |

## E  DETAILS ON COMPARED METHODS (SEC. 4)

Here, we give a slightly elaborated summary of the compared methods.

1. $\tau$-norm (Kang et al., 2019) is a two-stage decoupling method, which consists of classifier re-training(cRT) and learnable weight scaling(LWS).

2. BALMS(Ren et al., 2020) is also a two-stage method, where the balanced softmax loss and meta sampler is used in classifier training stage.

3. PaCo(Cui et al., 2021) uses supervised contrastive learning to learn the class centers in long-tailed data distributions.

4. MiSLAS(Zhong et al., 2021) focuses on calibrating the model properly on LT data with a two-stage method that uses CutMix(Yun et al., 2019) to enhance the representation learning stage. The classifier learning stage consists of label-aware smoothing and shift learning on batch normalization layers.

5. DiVE(He et al., 2021) use knowledge distillation where the teacher network is trained by existing well perfroming LT-methods such as BALMS and then trains student model with KL divergence loss and balanced softmax loss.

## F  DETAILS ON THE EXTENDED LT BENCHMARK (SEC. 4)

The newly added LT datasets are derived from existing balanced (*i.e.*, non-LT) computer vision datasets. They include Caltech101 (Li et al., 2022), CUB-200-2011 (Wah et al., 2011), Stanford Dogs (Khosla et al., 2011), Stanford Cars (Krause et al., 2013), DTD (Cimpoi et al., 2014), FGVC-Aircraft (Maji et al., 2013), Flowers-102 (Nilsback & Zisserman, 2008), and Fruits-360 (Mureşan & Oltean, 2017). We first sort the class indices with the number of samples per class to create a class order. We then set the imbalance ratio and sub-sample the data in each of the classes. In the sub-sampling, we use an exponential decaying curve to determine the class sample frequency from head to tail classes. As the number of data in each of the classes is limited, we use two popular imbalance ratios such as 10 or 100.

Further, we detail the dataset statistics for all the LT datasets used in our experiments in Tab. 6, including the newly added ones and the previously used datasets in the LT literature. The added datasets have a wide variety of semantic domains which may have not been covered by the existing LT datasets in the literature.

Table 7: Accuracy (%) of the proposed CANDLE with different teacher networks. CANDLE uses the ImageNet1k teacher as detailed in Sec. 4 and CANDLE$^{\ddagger}$ uses the ImageNet21K teacher which was originally used for large-scale experiments in Tab. 2.

| Datasets (Imb. Ratio) | **CANDLE** (ImageNet1K Teacher) | **CANDLE**$^{\ddagger}$ (ImageNet21K Teacher) |
|---|---|---|
| Caltech101 (10) | 55.31 | 52.27 |
| CIFAR-10 (10) | 91.76 | 91.27 |
| CIFAR-10 (100) | 85.01 | 85.59 |
| CIFAR-100 (10) | 66.09 | 64.93 |
| CIFAR-100 (100) | 50.88 | 53.15 |
| CUB-200-2011 (10) | 42.94 | 49.45 |
| Stanford Dogs (10) | 58.79 | 63.51 |
| Stanford Cars (10) | 51.06 | 49.73 |
| DTD (10) | 38.56 | 40.64 |
| FGVC-Aircraft (10) | 39.78 | 41.64 |
| Flowers-102 (10) | 64.61 | 66.27 |
| Fruits-360 (100) | 100.00 | 100.00 |
| Mean Acc. | 62.07 | 63.20 |

## G  More Discussion Regarding $\tau$-norm and BALMS (Sec. 4.2)

On comparison with existing LT methods on the small-scale datasets in Tab. 1, $\tau$-norm and BALMS seems to show relatively low accuracy. We hypothesize the following reasons for the low accuracy. For $\tau$-norm, it is based on how the weight norms distributed when the model is trained with the SGD optimizer. Since we use the Adam optimizer to train the models due to having binary backbones (Sec. 4), the core assumption for $\tau$-norm may not hold and hence the resulting low accuracy. For BALMS, it may have lacked the sufficient number of epochs to properly converge as we use binary backbone networks which usually take longer to train than floating point models (Courbariaux et al., 2015).

## H  Effect of Using Larger Teacher Networks for CANDLE (Sec. 4.2)

We used the ImageNet1k pretrained teacher network for small-scale experiments in Tab. 1 as we believe that the teacher network provides sufficient supervisory signals for the small-scale datasets. Nonetheless, it is interesting to utilize larger teacher network on the small-scale datasets. In particular, we use a teacher pretrained on ImageNet21k which was originally used only for the large-scale datasets.

As shown in Tab. 7, not all datasets benefit from the larger teacher network such as Caltech101 (10), CIFAR-10 (10), CIFAR-100 (10), and Stanford Cars (10). However, there are noticeable gains of $+6.51\%$ on CUB-200-2011-10 and $+4.72\%$ on Stanford Dogs-10 and the resulting mean accuracy is improved by $+1.13\%$. For the large-scale datasets in Tab. 2, we could not use the ImageNet1K teacher as that is a direct super set of the ImageNet-LT hence why we used the ImageNet21K teacher for the large-scale experiments. However, the gains in Tab. 7 suggest that one could use the ImageNet21K teacher for the small-scale datasets as well.

## I  Additional Comparison to TADE (Zhang et al., 2021b) (Sec. 4.2)

For a more comprehensive comparison, we also present results for TADE (Zhang et al., 2021b), which is an ensemble-based method. As shown in Tab. 8, we show results for TADE on the 12 small-scale datasets along with CANDLE. Our method, which uses a single model with no ensemble techniques, still outperforms TADE by large margins of $+24.94\%$ in terms of mean accuracy. Not only that, the proposed method performs well for both on the frequently used CIFAR-10 (10, 100) and CIFAR-100 (10, 100) datasets but also on the newly added datasets such as CUB-200-2011 (10), Stanford Dogs (10), and Stanford Cars (10).

Table 8: Accuracy (%) of the proposed CANDLE with TADE (Zhang et al., 2021b) which is an ensemble-based method. Our method shows superior performance against even ensemble-based TADE by large margins across many different datasets with a margin of $+24.94\%$ in mean accuracy.

| Dataset (Imb. Ratio) | TADE | **CANDLE** |
|---|---|---|
| Caltech101 (10) | 43.86 | **55.31** |
| CIFAR-10 (10) | 53.54 | **91.76** |
| CIFAR-10 (100) | 47.63 | **85.01** |
| CIFAR-100 (10) | 29.75 | **66.09** |
| CIFAR-100 (100) | 28.68 | **50.88** |
| CUB-200-2011 (10) | 13.00 | **42.94** |
| Stanford Dogs (10) | 17.20 | **58.79** |
| Stanford Cars (10) | 8.73 | **51.06** |
| DTD (10) | 27.98 | **38.56** |
| FGVC-Aircraft (10) | 12.69 | **39.78** |
| Flowers-102 (10) | 62.46 | **64.61** |
| Fruits-360 (100) | 100.00 | **100.00** |
| Mean Acc. | 37.13 | **62.07** |

Table 9: Average, head, medium, and tail class accuracy of CANDLE. On most datasets, CANDLE shows high tail class accuracy compared to the average, head, and medium class accuracy. Interestingly, this trend is stronger for large scale datasets where the tail class accuracy is higher or on par with the average accuracy.

| | Dataset | Average | Head | Medium | Tail |
|---|---|---|---|---|---|
| | Caltech101 (10) | 55.31 | 49.25 | 46.74 | 35.54 |
| | CIFAR-10 (10) | 91.76 | 95.50 | 90.50 | 91.03 |
| | CIFAR-10 (100) | 85.01 | 93.90 | 81.90 | 82.67 |
| | CIFAR-100 (10) | 66.09 | 75.64 | 67.20 | 54.00 |
| | CIFAR-100 (100) | 50.88 | 73.94 | 54.11 | 19.07 |
| Small Scale | CUB-200-2011 (10) | 42.96 | 50.94 | 40.13 | 26.30 |
| | Stanford Dogs (10) | 58.79 | 67.06 | 57.09 | 45.35 |
| | Stanford Cars (10) | 51.06 | 67.65 | 48.84 | 37.09 |
| | DTD (10) | 38.56 | 59.82 | 41.32 | 26.61 |
| | FGVC-Aircraft (10) | 39.78 | 29.55 | 51.05 | 42.04 |
| | Flowers-102 (10) | 64.61 | 91.39 | 81.11 | 41.33 |
| | Fruits-360 (100) | 100.00 | 100.00 | 100.00 | 100.00 |
| | Places-LT | 34.11 | 34.62 | 33.83 | 37.41 |
| Large Scale | ImageNet-LT | 49.10 | 53.89 | 42.21 | 45.66 |
| | iNat-LT | 43.53 | 48.94 | 41.96 | 46.35 |

## J   HEAD, MEDIAN, AND TAIL CLASS ACCURACY (SEC. 4.2)

To provide more information regarding the experimental results of CANDLE, we provide average, head, medium, and tail class accuracy of CANDLE for the 15 datasets in Tab. 9. For most of the datasets, CANDLE shows high accuracy at the tail classes compared to the average, head, or the medium classes. This is especially true for large-scale datasets such as Places-LT, ImageNet-LT, and iNat-LT where the tail class accuracy is either on par or higher than the average accuracy.

Additionally, in Fig. 8, 9 and 5 (in the main paper), we plot a per-class accuracy improvement of CANDLE over the baseline used in Tab. 3 following (Kozerawski et al., 2020) on Places-LT, ImageNet-LT, and iNat-LT, respectively. The figures show that CANDLE improves the accuracy at the tail and medium classes by large margins for all three datasets. Additionally, in the case of ImagetNet-LT, even the head accuracy shows noticeable improvements. The figures empirically show that CANDLE shows good average accuracy not because it is only good at head classes but mostly because it improves the accuracy of tail and medium classes by large margins.

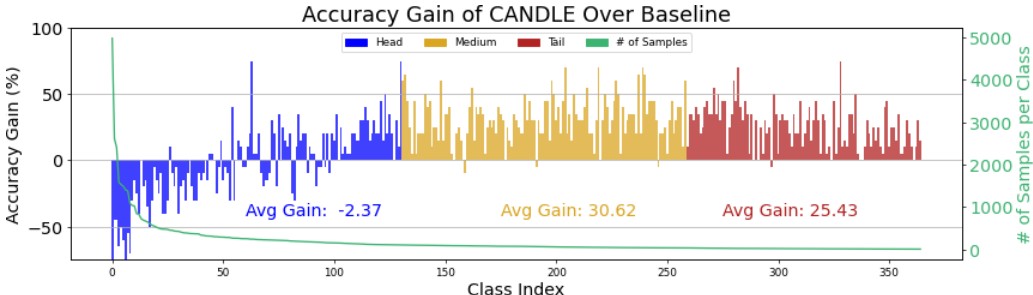

Figure 8: Per-class accuracy gain of CANDLE over the baseline on Places-LT. Our method improves the accuracy at the tail classes by $+25.43\%$, showing its effectiveness for LT. Additionally, the proposed method also improves the accuracy at the medium classes by $+30.62\%$ and only drops by $-2.37\%$ for the head classes.

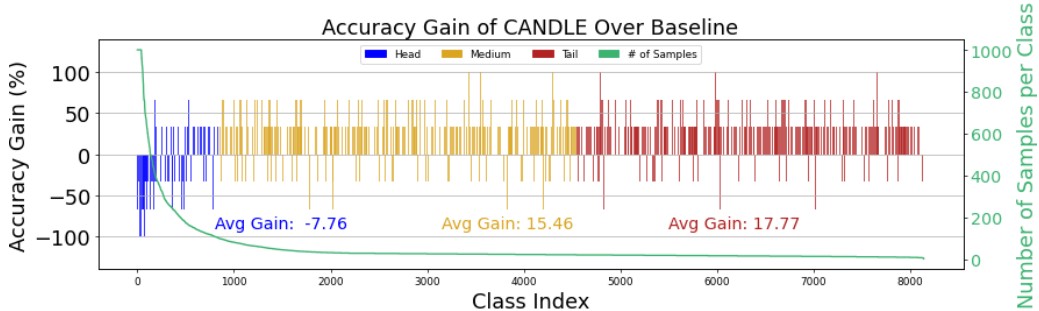

Figure 9: Per-class accuracy gain of CANDLE over the baseline on iNat-LT. Our method improves the accuracy at the tail classes by $+17.77\%$, showing its effectiveness for LT. The medium class accuracy is also improved by $+15.46\%$. There is a slight reduction in head class accuracy of $-7.76\%$.

## K  MORE DETAILED ABLATION STUDY RESULTS (SEC. 4.4

We presented the mean accuracy over the 12 small-scale datasets for our ablation studies in Tab. 3 (in the main paper) for brevity. We also report the accuracy on each of the 12 datasets in Tab. 10 to provide more details regarding our ablation studies. However, we also note that looking at the accuracy of a particular dataset only may not give the most accurate description.

Comparing ① to ① + ② + ③ (=CANDLE) in Tab. 3, 9 out of 12 datasets show an accuracy increase. On the 3 datasets where an accuracy decrease is observed, the accuracy drops by $-5.17\%$ at most, whereas on 9 datasets where the accuracy increases, it increases by up to $+41.16\%$. More interestingly, the mean accuracy consistently goes up when add more and more parts of CANDLE, as also shown in Tab. 3.

## L  ADDITIONAL ANALYSIS ON THE CLASS WEIGHT NORMS

We further analyze the classifier weight and bias norms for binary networks in LT conditions as an extended discussion from Sec. 4.1. As shown in Fig. 10-(a), we see the classifier weight norms differ in magnitude across classes when the binary network is trained from scratch. Looking at the bias in Fig. 10-(b), we see the classifier bias also having differing norms for different classes when the binary network is trained from scratch. In contrast, when CANDLE is applied instead, the substantial difference in the classifier weight and bias norms are mitigated, which implies that CANDLE stabilizes the norms of both the classifier weight and biases in binary networks.

Table 10: A more detailed ablation studies of our method CANDLE. The mean accuracy over 12 small-scale datasets as well as the accuracy for each dataset are shown. The circled numbers denote the following ①: 'Calibrate & Distill', ②: Data-Driven Bal., ③: Multi-Res., respectively. Using our 'Calibration & Distill' framework drastically improves the performance across all the tested datasets when compared to the baseline of using the pretrained weights without calibration. 'Data-Driven Bal.' and 'Multi-Res.' also improve the accuracy substantially over all the tested datasets.

| Datasets (Imb. Ratio) | Baseline | Ablated Models | | |
|---|---|---|---|---|
| | | ① | ① + ② | ① + ② + ③ (=CANDLE) |
| Caltech101 (10) | 36.75 | 53.44 | 53.37 | 55.31 |
| CIFAR-10 (10) | 88.90 | 82.08 | 92.34 | 91.76 |
| CIFAR-10 (100) | 70.00 | 43.85 | 87.24 | 85.01 |
| CIFAR-100 (10) | 48.96 | 69.07 | 68.58 | 66.09 |
| CIFAR-100 (100) | 33.62 | 54.92 | 53.70 | 50.88 |
| CUB-200-2011 (10) | 12.69 | 37.66 | 37.61 | 42.94 |
| Stanford Dogs (10) | 25.66 | 63.96 | 64.42 | 58.79 |
| Stanford Cars (10) | 25.87 | 44.00 | 45.19 | 51.06 |
| DTD (10) | 21.33 | 40.21 | 36.38 | 38.56 |
| FGVC-Aircraft (10) | 28.02 | 29.91 | 29.22 | 39.78 |
| Flowers-102 (10) | 51.86 | 62.16 | 63.82 | 64.61 |
| Fruits-360 (100) | 99.61 | 85.85 | 100.00 | 100.00 |
| Mean Acc. | 45.27 | 55.59 | 60.99 | 62.07 |

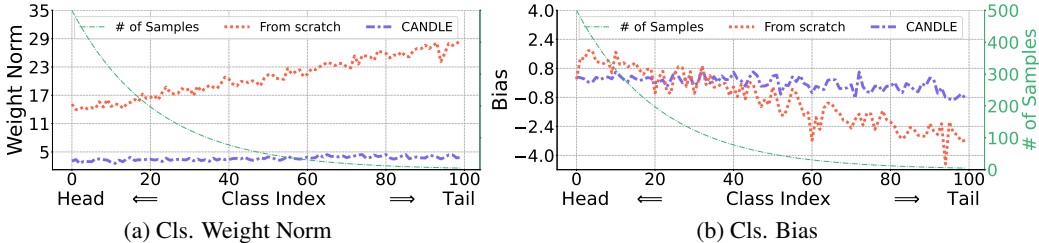

(a) Cls. Weight Norm          (b) Cls. Bias

Figure 10: Classifier weight norm and bias for binary networks trained from scratch or using CANDLE on CIFAR-100 (100). We can see that training from scratch results in binary networks with large magnitude differences across classes for both weights and biases whereas using CANDLE, binary networks have relatively low such difference for them instead.

