# OpenReview forum: "Long-Tailed Recognition on Binary Networks by Calibrating A Pre-trained Model"
_ICLR.cc/2024/Conference — Submitted to ICLR 2024_

### Official Review · Reviewer_HnuF · 2023-10-31

**Soundness:** 3 good
**Presentation:** 3 good
**Contribution:** 3 good
**Rating:** 6
**Confidence:** 3

**Summary:**

The paper proposes to train a binary network for long-tail recognition with the Calibrate and Distill framework, which fine-tunes the pre-trained model and trains the binary network at the same time with an adversarial balancing scheme.

**Strengths:**

1. The paper is clearly written.

2. The long-tail recognition problem in the low computation resource is quite relevant for machine learning applications.

3. The empirical study is intensive, which tests the proposed method and 6 baselines on 15 datasets.

4. The effectiveness of the proposed method is quite impressive.

**Weaknesses:**

The proposed method contains an MLP for learning the weighting hyperparameter. It is unclear how the architecture and the training hyperparameters of the adversarial network affect the final performance. More generally, the proposed method contains the training of three models and each of them may need difference training hyperparameters, which may make the hyperparameter tuning cumbersome.

Minor: Fig. 8 and 9 are not quite clear as they are not high-resolution.

**Questions:**

Algorithm 1 shows that the learning rate for the three components is the same. Is this consistent with the experimental setting?

---

> ### Author Response · Authors · 2023-11-17
> **Author's reponse to reviewer HnuF**
>
> > It is unclear how the architecture and the training hyperparameters of the adversarial network affect the final performance. More generally, the proposed method contains the training of three models and each of them may need difference training hyperparameters, which may make the hyperparameter tuning cumbersome.
>
> $\to$ Architecture of the MLP for $\lambda_\phi$: The MLP has 3 layers with the first layer having 2 channels (as the input is two loss values), and the second and third layers have 32 and 512 channels respectively. The final weighting parameter (as we learn this, it is rather a parameter than a hyper-parameter) is obtained by averaging out the third layer output and passing it to a sigmoid function. We will add this detail in the revision.
>
> $\to$ Training hyper-parameters: we found that using the same learning for the 3 components in Alg 1. yields robust results as opposed to tuning the hyperparameters. This is due to the MLP being relatively simple in architecture which makes it easier to train robustly. Then, the MLP re-weights the loss values adaptively, which makes training the other component more stable.
>
> $\to$ Effect on performance: We experimented with variations of the architecture and hyperparameters of the MLP but it had little impact on the final performance.
>
> > Minor: Fig. 8 and 9 are not quite clear as they are not high-resolution.
>
> $\to$ Thank you for the suggestion! We will change them to high resolution in the revision.

---

> ### Author Response · Authors · 2023-11-22
> **Discussion reminder**
>
> We sincerely thank you for your effort in reviewing our submission. We gently remind the reviewer that we tried our best to address your concerns by our replies. As the discussion period is nearing the end, we would be delighted to hear more from you if there are any further concerns.

---

> > ### Author Response · Authors · 2023-11-23
> > **Revision upload**
> >
> > We remind the review that we uploaded a revised version of our submission that addresses some of the reviewer's concern.
> > Notably, the details on the architecture of the MLP can be found in Sec. D of the appendix.

---

### Official Review · Reviewer_udmW · 2023-10-31

**Soundness:** 3 good
**Presentation:** 2 fair
**Contribution:** 3 good
**Rating:** 6
**Confidence:** 4

**Summary:**

This paper addresses the long-tailed distribution in datasets on the training of efficient binary neural networks. By calibrating classifier layers of a large pre-trained teacher model to long-tailed datasets, this paper introduces a novel calibrate-and-distill framework. This method allows binary models to be distilled on the balanced pre-trained teacher. To further enhance generalization, the authors incorporate adversarial balancing and employ multi-resolution input techniques. Extensive experiments are conducted to validate the efficacy of the proposed CANDLE methods.

**Strengths:**

1. The constructed challenge of LT problem in binary NN directly relates to real-world applications, making it a compelling research question.
2. The design of the proposed "calibrate-and-distill" framework is both logical and coherent, adeptly aligning the general visual capabilities of teacher models with LT data. The detailed implementation makes it convincing.
3. The performance of CANDLE significantly outperforms that of other methods.

**Weaknesses:**

1. Although LT can pose challenges in binary NN training, it would greatly benefit the reader if there were experiments, illustrative demonstrations, and/or theoretical analysis that elucidate the specific difficulties it presents.

2. There are some instances where the writing or logical flow could be improved. I've noted some specific cases below:

2.1. The Introduction mentions various instances of large-scale generative models. However, given that the core technology of this paper focuses on discriminative tasks, it might be more relevant to provide examples like DINO and MAE.

2.2. The motivation part of the Introduction has the same problem.

2.3. As the framework is around the concept of "calibrate-and-distill", it would be beneficial to provide some context on domain adaptation and model distillation in the Related Work section.

2.4 In the Approach section, clear mathematical formulations for the LT-aware CE loss and feature similarity are essential. Not all readers might be as familiar with these concepts as us.

2.5 The implementation of Multi-resolution learning in relation to the LT problem isn't well-defined. Elaborating on this would provide clarity.

**Questions:**

1. Regarding section 3.3, it appears that multi-resolution learning is primarily for efficient training. Would it be more fitting to discuss this within the implementation of experiments? Alternatively, is it integral to the overall framework's effectiveness?

---

> ### Author Response · Authors · 2023-11-17
> **Author's reponse to reviewer udmW**
>
> > Special difficulties for binary LT: Although LT can pose challenges in binary NN training, it would greatly benefit the reader if there were experiments, illustrative demonstrations, and/or theoretical analysis that elucidate the specific difficulties it presents.
>
> $\to$ Theoretically speaking, as binary networks binarize the weights to 1 and -1 depending on a threshold, small differences around the threshold results in high variation. In other words, parameter updates near the threshold are exacerbated which means a few samples can incur large changes in the binary weights. Unfortunately, the changes in the binary weights is what actually trains the binary network even if they are exacerbated. Thus, gradient updates from samples in the long-tail classes could be amplified, which would worsen the overfitting to the small number of training samples in the LT class. In other words, the binary networks are more prone to data scarcity in the LT data.
> As an illustrative empirical demonstration, we present Fig. 4-(a) that shows that the classifier weight norms for tail classes rose more for binary networks than for FP networks. Higher classifier weight norms for tail classes indicates the network learned to multiply high weights to features present in the infrequent training samples in the long-tail classes.
>
>
> > Flow improvement: There are some instances where the writing or logical flow could be improved. I've noted some specific cases below:
>
> $\to$ Thank you for the valuable suggestions to improve 2.1 2.2, 2.3 and 2.4. We will change them accordingly in the revision.
>
> For your question about Sec. 2.5: Multi-resolution training can be understood as data augmentation where we explicitly obtain more training data by using differently resized images. This is particularly beneficial for our problem of binary LT as binary LT is more prone to the data-scarcity in long-tailed classes (please refer to our answer regarding the question on special difficulties of binary LT ).
>
> > Regarding section 3.3, it appears that multi-resolution learning is primarily for efficient training. Would it be more fitting to discuss this within the implementation of experiments? Alternatively, is it integral to the overall framework's effectiveness?
>
> $\to$ We agree that the implementation of the multi-resolution could be discussed within the experiment section. Rather, we should provide the motivation for the multi-scale training in the approach instead, as described in our second answer to the question regarding flow improvement in Sec. 2.5. While it is not integral to our approach, it certainly does help when we can utilize it (please see Tab 3 for ablation results).

---

> > ### Comment · Reviewer_udmW · 2023-11-22
> >
> > Thank you for your explanations. I don't have any further questions at present. I will give my final review accordingly.

---

### Official Review · Reviewer_eG2e · 2023-11-01

**Soundness:** 2 fair
**Presentation:** 3 good
**Contribution:** 2 fair
**Rating:** 3
**Confidence:** 4

**Summary:**

This paper proposes to train the binary networks on the long tailed dataset. The motivation of the paper is to train the resource constrained binary network on a real-world long-taile dataset. For this the authors propose the Calibrate and Distill Framework, which uses a combination of feature similarity loss and logit distillation loss, to jointly optimize the neural network on the long tailed data. The three major components of the proposed approach are calibrate and distill, Adversarial lambda learning and Multi-Resolution training. The effectiveness of the proposed approach is demonstrated across 15 datasets including both the small scale and large scale dataset.

**Strengths:**

New Problem Introduced for the community.

Paper is well written and clear, with sufficient experimental results.

**Weaknesses:**

Motivation: I am unable to understand why this method is tailored towards long-tailed data. The loss functions proposed are generic and are not tailored towards long-tailed data.

Baselines: The baselines used in the current study are LT baselines which are not suitable to be directly applied for binary networks, hence perform inferiorly for binary networks. Creation of fair baselines where a reasonable method for training binary networks should be combined with LT methods. Hence, the current comparisons provided are unfair and require improvement. Some approaches from https://arxiv.org/pdf/2110.08562.pdf could be handy.

Missing References: Recently some works have been proposed which use flatness in long-tail learning which can be used for learning quantized networks on LT data. See [R1, R2, R3].

Setup: The pre-trained model used for distillation is trained on some other dataset, and hence doesn’t posses the same LT distribution as on target dataset. Hence, it’s unclear whether the LT classes in target dataset really constitute the long tail distribution or not.

[R1]: Foret, Pierre, et al. "Sharpness-aware minimization for efficiently improving generalization." arXiv preprint arXiv:2010.01412 (2020).

[R2]: Na, Clara, Sanket Vaibhav Mehta, and Emma Strubell. "Train flat, then compress: Sharpness-aware minimization learns more compressible models." arXiv preprint arXiv:2205.12694 (2022).

[R3]: Rangwani, Harsh, Sumukh K. Aithal, and Mayank Mishra. "Escaping saddle points for effective generalization on class-imbalanced data." Advances in Neural Information Processing Systems 35 (2022): 22791-22805.

**Questions:**

Why its a long tailed method? The multi-scale training, calibration and distillation and adversarial loss balancing all can be perfectly used on the balanced dataset. Can the authors please provide explanation regarding?

---

> ### Author Response · Authors · 2023-11-17
> **Author's reponse to reviewer eG2e**
>
> > Setup: The pre-trained model used for distillation is trained on some other dataset, and hence doesn’t possess the same LT distribution as on target dataset. Hence, it’s unclear whether the LT classes in target dataset really constitute the long tail distribution or not.
>
> $\to$ We cordially argue that whether the LT classes in the target dataset constitute the long tail distribution in the pretraining data is not a critical matter to our proposed approach. Specifically, we design our method to not be dependent on how close or far the pretraining data and the target LT data might be as follows. The  calibration process improves the pretrained model on the LT dataset by learning a newly attached classifier. The distillation supervision from the frozen feature vectors and the classifier logits are adversarially balanced in a data-driven manner. Thus, if the pretraining data covers the target LT classes, our distillation emphasizes more on frozen feature vector supervision and vice versa, all without needing human tweaking. This design choice allows us to use a wide array of LT datasets (15 in total) with the **same** methodology and still achieve superior performance.
>
> > Motivation: Why is this a long-tailed method?
>
> $\to$ It is because the calibrate part of the proposed calibrate and distillation is used specifically so that the distillation stage can work whether the pretraining data covers the target LT classes or not (please refer to our answer on the question regarding Setup). In addition, the multi-scale training is more effective for binary LT as binary LT is more prone to the data-scarcity in long-tailed classes (please refer to our answer to reviewer udmW on question A). We will revise the manuscript to better reflect on this in the revision soon.
>
> Perhaps more importantly, as this is a new (eG2e), compelling (udmW), and relevant (HnuF) problem for the community, we strived to introduce a general framework which showed strong performance (udmW, HnuF) but was open to modifications down the road.
>
> > Baselines
>
> $\to$ We cordially argue that as there was no prior work on binary LT, we did our best of using the same binary backbone architecture (which already incorporates some of the techniques mentioned in the pointed paper) for all compared methods and ours.
>
> In addition, for certain instances where the performance was too low (*e.g.*, DiVE with binary networks), we swapped out the part that was causing the low performance with some of our proposed components (*e.g.*, the teacher network part in DiVE) and created a better baseline (*e.g.*, DiVE*) which we also compare with (please see Table 1 & 2).
>
> > Missing references:
>
> $\to$ We appreciate the reviewer for pointing out the work. We will review them in the related works section in the revision and incorporate some of the mentioned techniques into our calibrate & distill framework.

---

> ### Author Response · Authors · 2023-11-22
> **Discussion reminder**
>
> We sincerely thank you for your effort in reviewing our submission. We gently remind the reviewer that we tried our best to address your concerns by our replies. As the discussion period is nearing the end, we would be delighted to hear more from you if there are any further concerns.

---

> ### Author Response · Authors · 2023-11-23
> **Revision upload**
>
> We remind the review that we uploaded a revised version of our submission that addresses some of the reviewer's concern.
> Notably, we added more description on why the proposed method is tailored towards the long-tail setting along with discussion on the pointed out related works.

---

> ### Comment · Reviewer_eG2e · 2023-12-03
> **Response to The Authors**
>
> Thanks for the rebuttal. I want to highlight my concerns, which I still feel are unaddressed:
>
> 1. Usage of Non-LT pre-trained model. In the long-tail setup, it is assumed that the head class knowledge is transferred to the tail class, making it an exciting problem distinct from the transfer learning setup. Hence, I am still unconvinced about using any pre-trained model in the LT setup.
>
> 2. Missing Baselines: I would like to see some baselines coming from the literature on training binary networks. Just naively using LT methods on binary networks is potentially not a great idea. Hence, it would be great to have more baselines going further.
>
> Hence, I will keep my score.

---

### Official Review · Reviewer_fEHn · 2023-11-01

**Soundness:** 3 good
**Presentation:** 3 good
**Contribution:** 3 good
**Rating:** 3
**Confidence:** 3

**Summary:**

The authors introduce a "calibrate-and-distill" framework that leverages pretrained full-precision models on balanced datasets as teachers for distilling knowledge into binary networks on long-tailed datasets. They also propose an adversarial balancing mechanism and a multi-resolution learning scheme for improved generalization. Empirical validation on 15 datasets, including newly created long-tailed datasets, demonstrates performance improvements over previous methods.

**Strengths:**

- Using a large pre-trained network trained on the other dataset is a novel approach in long-tailed recognition.
- Plenty of experimental results is provided helping with understanding.
- The paper is well-writen and easy to follow

**Weaknesses:**

- This paper demonstrates limited novelty, as it employs established methodologies, including knowledge distillation, multi-resolution techniques, adversarial training, and binary networks, which are well-documented in the existing literature.
- The scalability of this approach raises concerns, given its utilization of a teacher model of substantial size. Notably, in Table 2, only single-resolution is utilized, primarily due to VRAM limits.
- The incorporation of a non-LT pretrained network in advance does not appear to be a practical approach. One of the underlying motivations for addressing the long-tailed recognition problem stems from the inherent imbalances frequently encountered in natural datasets.
- It is advisable to update the set of compared methods in the experimental evaluation, as the current selection appears somewhat outdated. A more comprehensive comparison with contemporary methodologies is suggested.
- The mechanism by which the parameter $\lambda_\phi$ facilitates generalization across multiple data distributions remains unclear. In Figure 6, it is evident that, across all datasets, $\lambda_\phi$ consistently converges to a value of 1, indicating that towards the conclusion of the training process, improvements are predominantly focused on the classifier rather than the encoder. If the variation in the scheduling of $\lambda_\phi$ is considered essential for enhancing generalization, a more thorough explanation is needed.

**Questions:**

- How the large teacher model can effectively contribute to the performance of the binary network, particularly when a substantial domain gap exists between the dataset used for pretraining, such as ImageNet, and the actual target dataset, which could be, for instance, an MRI dataset in practical applications.

---

> ### Author Response · Authors · 2023-11-17
> **Author's reponse to reviewer fEHn**
>
> > Limited novelty
>
> $\to$ While we definitely agree that we are standing on the shoulders of giants, we argue that the calibrate and distill framework which uses pretrained models in the context of LT (or binary LT) is novel (fEHn).  In addition, the way we use the knowledge distillation (KD) is different from most of the KD methods (*e.g.”, distilling feature extraction and classifier information with learnable balancing parameters) with application of adversarial learning on the balancing parameters being novel. Lastly, learning multi-resolution inputs in the proposed way is novel and computationally efficient.
> Orthogonally, our contribution also lies in introducing a new and important problem (eG2e, udmW, HnuF) to the community, whilst also providing a simple yet strong baseline (udmW, HnuF) that is void of bells-and-whistles for ease of modification to the research community.
>
> > Scalability concern, given its utilization of a teacher model of substantial size. Notably, in Table 2, only single-resolution is utilized, primarily due to VRAM limits.
>
> $\to$ We cordially argue that the scalability concern is not critical for multiple reasons. First, we can simply calculate the teacher model’s output on the dataset beforehand which will substantially reduce VRAM overhead. Second, the multi-scale training is not integral to our calibrate and distill framework. If we did not have enough VRAM to utilize it, we could simply forgo it and still obtain good performance. Last, at the rates in which VRAM size is advancing with today’s hardware improvement, we believe that the size of our teacher models are not that limiting with regards to VRAM.
>
> >  Practicality of incorporating non-LT pretrained models in advance.
>
> $\to$ As great efforts have been devoted into curating publicly available large-scale datasets that are balanced such as ImageNet-1k and ImageNet-21K, we argue that obtaining non-LT pretrained models in advance is indeed not difficult, $i.e.*, quite practical. We also note that we do **NOT** assume that a pretrained model is trained on either LT or non-LT distributions. Instead, we utilize a cheap calibration process (training only a linear classifier) for the pretrained model, which makes it possible to utilize the pretrained model as a distillation teacher no matter what the pretrained data distribution was. Empirically, the calibration noticeably improves the performance by more than 10% compared to using a non-calibrated teacher (please see Tab 3 for the ablation studies).
>
> > Update the compared methods.
>
> $\to$ We will definitely update them for the final version. But we believe that the strong performance of CANDLE,  as noted by other reviewers (udmW, Hnun, HnuF) as well,  is enough to provide empirical validations..
>
> > Mechanism of \lambda_phi. In Figure 6, it is evident that, across all datasets, \lambda_phi consistently converges to a value of 1, indicating that towards the conclusion of the training process, improvements are predominantly focused on the classifier rather than the encoder. If the variation in the scheduling of \lambda_phi is considered essential for enhancing generalization, a more thorough explanation is needed.
>
> $\to$ As $\lambda_\phi$ approaches the value of 1, the binary encoder is primarily trained with the supervision from the pretrained model’s feature extractor (please see Eq. 4), which is inline with the reviewer’s comment and also mentioned in Sec. 4.3 of the paper.  However, as pointed out, the scheduling of the $\lambda_\phi$ is important for generalization as it varies substantially from dataset to dataset. Our formulation of $\lambda_\phi$ allows a schedule to be learned solely by the target LT data and not by humans. This makes it applicable to multiple target LT datasets without the need for human intervention.
>
> > How large teachers help when there is a significant domain gap.
>
> $\to$ Thank you for the interesting question. Currently, the teacher models with the size as used in our work perform well for addressing domain gaps such as from ImageNet to Places dataset. While we are also curious about the effect of teacher model size on the performance of addressing domain gaps, the experimental undertaking is not trivial for two reasons. First, it is difficult to quantify the domain gap, which makes plotting the teacher size vs its effect on addressing the domain gap hard. Second, isolating the effect of teacher size scaling is hard as the architectural differences of differently sized models also have an effect. Thus, we leave this as an interesting future research avenue.
>
> Meanwhile, to better address larger domain gaps, we believe that improving the calibration process to better handle the large domain gaps given the same model size budget also could be another interesting future work (*i.e.*, gaining hints from the transfer learning and domain adaptation literature).

---

> ### Author Response · Authors · 2023-11-22
> **Discussion reminder**
>
> We sincerely thank you for your effort in reviewing our submission. We gently remind the reviewer that we tried our best to address your concerns by our replies. As the discussion period is nearing the end, we would be delighted to hear more from you if there are any further concerns.

---

### Author Response · Authors · 2023-11-22
**Author response reminder**

We appreciate the valuable feedback from the reviewers. Please note that individual responses to the reviewers were posted and we would love to hear back.

---

### Meta-Review · Area_Chair_9iPT · 2023-12-19

**Metareview:**

The paper proposes the challenge of learning long-tailed (LT) distributions under resource constraints. The authors explore binary neural networks as the main backbone to address the resource constraints, and propose a distillation framework from a pretrained full-precision model for LT classes. While the reviewers were generally positive about LT learning and binary neural networks, they raised multiple concerns that were viewed by AC as critical issues:
1) significance of the combination of the two (binary networks and distillation from a model pre-trained on non-LT data) as the main contribution - see Reviewer eG2e and Reviewer udmW comments about (i) lack of motivation of the setting, and (ii) lack of evidence that this is indeed LT method and setting;
2) insufficient empirical evidence to evaluate that such combination is indeed addressing the challenge in the best possible way  - (i) see Reviewer eG2e concerns about unfair baseline comparisons, and missing baselines that define the scope of contributions, and (ii) reviewer fEHn concern about substantial domain gap between the dataset used for pretraining, and the actual target dataset – studying this aspect could help to demonstrate the benefits of the proposed methodology, (iii) Reviewer HnuF and Reviewer fEHn concerns about hyper-parameter tuning.

The reviewers also express concerns about lack of motivation, limited novelty,  and parameter tuning, which were addressed in the rebuttal to some extent. The rebuttal was able to clarify some questions, but did not manage to sway any of the reviewers. In light of an unanimous lack of enthusiasm for this work, a general consensus among reviewers and AC was reached to reject the paper. We hope the reviews are useful for improving and revising the paper.

**Justification For Why Not Higher Score:**

The rebuttal was able to clarify some questions, but did not manage to sway any of the reviewers. In light of an unanimous lack of enthusiasm for this work, a general consensus among reviewers and AC was reached to reject the paper.

**Justification For Why Not Lower Score:**

N/A

---

### Decision · Program_Chairs · 2024-01-16

Reject